# On Task-personalized Multimodal Few-shot Learning for Visually-rich Document Entity Retrieval

**Jiayi Chen**[1*]**, Hanjun Dai**[4]**, Bo Dai**[2,4]**, Aidong Zhang**[1]**, Wei Wei**[3,4◇]

[1]University of Virginia, [2]Georgia Institute of Technology, [3]Accenture
[4]Google, Mountain View, USA

{jc4td,aidong}@virginia.edu, {hadai,bodai}@google.com, wei.h.wei@accenture.com

## Abstract

Visually-rich document entity retrieval (VDER), which extracts key information (e.g. date, address) from document images like invoices and receipts, has become an important topic in industrial NLP applications. The emergence of new document types at a constant pace, each with its unique entity types, presents a unique challenge: many documents contain unseen entity types that occur only a couple of times. Addressing this challenge requires models to have the ability of learning entities in a *few-shot* manner. However, prior works for Few-shot VDER mainly address the problem at the document level with a predefined global entity space, which doesn't account for the *entity-level* few-shot scenario: target entity types are *locally personalized* by each task and entity occurrences vary significantly among documents. To address this unexplored scenario, this paper studies a novel *entity-level* few-shot VDER task. The challenges lie in the uniqueness of the label space for each task and the increased complexity of out-of-distribution (OOD) contents. To tackle this novel task, we present a task-aware meta-learning based framework, with a central focus on achieving effective *task personalization* that distinguishes between in-task and out-of-task distribution. Specifically, we adopt a hierarchical decoder (HC) and employ contrastive learning (ContrastProtoNet) to achieve this goal. Furthermore, we introduce a new dataset, FewVEX, to boost future research in the field of entity-level few-shot VDER. Experimental results demonstrate our approaches significantly improve the robustness of popular meta-learning baselines.

## 1 Introduction

Visually-rich Document Understanding (VrDU) aims to analyze scanned documents composed of structured and organized information. As a sub-problem of VrDU, the goal of Visually-rich Document Entity Retrieval (VDER) is to extract key information (e.g., date, address, signatures) from the document images such as invoices and receipts with complementary multimodal information (Xu et al., 2021; Garncarek et al., 2021; Lee et al., 2022). In real-world VDER systems, new document types continuously emerge at a constant pace, each with its unique *entity spaces* (i.e., the set of entity categories that we are going to extract from the document). This poses a substantial challenge: a large amount of documents lack sufficient annotations for their unique entity types, which is referred to as *few-shot entities*. To tackle this, Few-shot Visually-rich Document Entity Retrieval (FVDER) has become a crucial research topic.

Despite the importance of FVDER, there has been limited amount of prior works in this area. Recent efforts have leveraged pre-trained language models (Wang and Shang, 2022) or prompt mechanisms (Wang et al., 2023b) to obtain transferable knowledge from a source domain and apply it to a target domain, where a small number of document images are labeled for fine-tuning. These prior works address the few-shot problem in a granularity of **document level**, assuming a *globally predefined* entity space and *balanced* entity occurrences across documents. However, in certain real-world scenarios, the few-shot challenge can also manifest at the **entity level**–i.e., the number of entity occurrences in labeled documents are limited, assuming the situations where entity classes might be *locally specialized* by each user (task) and their occurrences maintain a significant *imbalance* across documents. In such scenarios, prior methods struggle to **(1)** efficiently address the model personalization on each task-specific label space and **(2)** effectively handle the increased complexity of out-of-distribution contents, with few-shot entity annotations.

To provide a complementary research perspec-

---

*Work done when Jiayi Chen interned at Google.
◇Corresponding author

| Method | Instance Granularity | Unseen Entities? | Entity Space | Entity Occurrence |
|---|---|---|---|---|
| (Wang and Shang, 2022) | document level | yes | globalized | balanced |
| (Wang et al., 2023b) | document level | no | globalized | balanced |
| Ours | entity level | yes | personalized | imbalanced |

Table 1: Comparison on task formulations and application scenarios.

tive alongside the existing document-level work, in this paper, we initiate the investigation for the unexplored **entity-level few-shot VDER**. To begin with, we *formulate* an $N$-way soft-$K$-shot VDER task setting together with a distribution of individual tasks, which simulates the application scenario of such task, that is, the user or annotator of each few-shot task is only interested in $N$ personalized entity types and the number of labelled entity occurrences in a task is within a flexible range determined by $K$ shots. Table 1 summarizes the differences in application scenarios between prior works and ours.

Then, to tackle the limitations of prior methods on this new task, we adopt a meta-learning based framework build upon pretrained language models, along with several proposed techniques for achieving *task personalization* and handling *out-of-task distribution* contents. With the help of the meta-learning paradigm, **(1)** the learning experiences on some example tasks could be effectively utilized and **(2)** the domain gap between the pre-trained model and novel FVDER tasks is largely reduced, promoting quicker and more effective fine-tuning on future novel entity types. Yet we found popular meta-learning algorithms (Finn et al., 2017; Snell et al., 2017; Chen et al., 2021) are still not robust to the entity-level $N$-way soft-$K$-shot VDER tasks. The difficulty is that the background context that does not belong to the task-personalized entity types occupies most of the predictive efforts, and also, such noisy contextual information varies a lot across tasks and documents. To address this, we propose **task-aware meta-learning** techniques (ContrastProtoNet, ANIL+HC, etc.) to allow the meta-learners to be aware of those multi-mode contextual out-of-task distribution and achieve fast adaptation to the task-personalized entity types.

Furthermore, we present a **new dataset**, named FewVEX, which comprises thousands of entity-level $N$-way soft-$K$-shot VDER tasks. We also introduce an automatic dataset generation algorithm, XDR, designed to facilitate future improvement of FewVEX, such as the expansion of the number of document types and entity types. Specifically, we set an upper bound for entity occurrences and

sample across the training documents in a way that guarantees to return soft-balanced few-shot annotations. Training documents are constructed in a way such that they cooperatively contain a certain number of entity occurrences per information type.

Our contributions are summarized as follows. **(1)** To the best of our knowledge, this paper is the first attempt studying the Few-shot VDER problem at the *entity level*, providing a complementary research perspective in addition to the existing document-level works. **(2)** We propose a meta-learning based framework for solving the newly introduced task. While vanilla meta-learning approaches have limitations on this task, we propose several task-aware meta-learners to enhance task personalization by dealing with out-of-task distribution. **(3)** Experiment results on FewVEX demonstrate our proposed approaches significantly improve the performance of baseline methods.

## 2 Entity-level Few-shot VDER Setting

**General VDER.** A document image is processed through Optical Character Recognition (OCR) (Chaudhuri et al., 2017) to form a sequence of tokens $X = [\mathbf{x}_1, \mathbf{x}_2, \ldots, \mathbf{x}_L]$, where $L$ is the sequence length and each token $\mathbf{x}_l$ is composed of multiple modalities $\mathbf{x}_l = \{\mathbf{x}_l^{(v)}, \mathbf{x}_l^{(p)}, \mathbf{x}_l^{(b)}, \ldots\}$ such as the token id $(v)$, the 1d position $(p)$ of the token in the sequence, the bounding box $(b)$ representing the token's relative 2d position, scale in the image, and so on. The goal is to predict $Y = [y_1, y_2, \ldots, y_L]$, which assigns each token $\mathbf{x}_l$ a label $y_l$ to indicate either the token is one of entities in a set of *predefined* entity types or does not belong to any entity (denoted as 0 class).

$N$**-way Soft-**$K$**-shot VDER in Entity Level.** In real-world Few-shot VDER systems with label scarcity, individual users often show their *personal interests* in a *small* number of *new* entity types. Such user-dependent **task personalization** gives rise to a novel **entity-level** task formulation of Few-shot VDER. Formally, an entity-level $N$-way soft-$K$-shot VDER task $\mathcal{T} = \{S, Q, \mathcal{E}\}$ consists of a train (*support*) set $S$ containing $M_s$ documents, a test (*query*) set $Q$ containing $M_q$ documents, and a

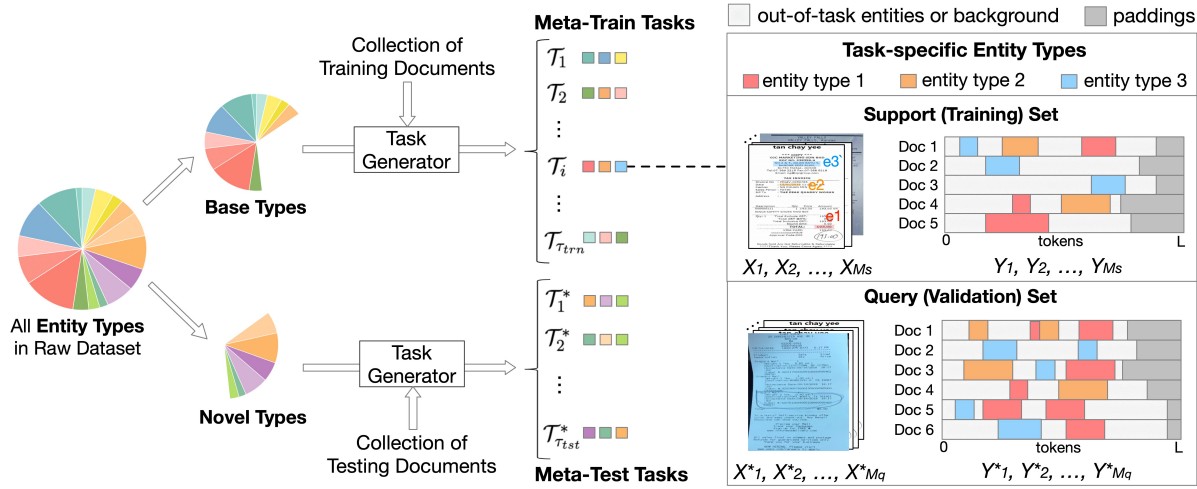

Figure 1: Proposed task formulation and problem setting. Different colors represent different entity types. The pie chart split on the left indicates that the target classes in testing tasks are not seen in training tasks. On the right area, we show an example 3-way soft-2-shot task. In this example, $\rho = 2$.

*target* class set $\mathcal{E}$ containing $N$ target entity types

$$S = \{(X_1, Y_1), \ldots, (X_{M_s}, Y_{M_s})\}$$
$$Q = \{X_1^*, X_2^*, \ldots, X_{M_q}^*\} \quad (1)$$
$$\mathcal{E} = \{e_1, e_2, \ldots, e_N\},$$

where $X_j = [\mathbf{x}_{j1}, \mathbf{x}_{j2}, ..., \mathbf{x}_{jL}]$ is the sequence of multimodal token features of document $j$, $Y_j = [y_{j1}, y_{j2}, ..., y_{jL}]$ is the sequence of token labels corresponding to $X_j$, and $e_c$ denotes the $c$-th entity type in $\mathcal{T}$. "$N$-**way**" refers to the $N$ unique entity types the user is interested in, reflecting task personalization. It is important to highlight that within $S$ and $Q$ documents, there may exist entities that fall outside the $N$ target classes ($e' \notin \mathcal{E}$). These entity types come from the *out-of-distribution* in contrast to what the task $\mathcal{T}$ aims to train on, which do not attract user interest, remain unlabeled, and thus are treated as the background 0 class. "**Soft-$K$-shot**" refers that, among the $M_s$ labelled documents in $S$, the total number of *occurrences* of each entity type $e \in \mathcal{E}$ is within a range $K \sim \rho K$, where $\rho > 1$ is the softening hyperparameter. An *entity occurrence* is defined as a contiguous subsequence in the document with the same entity type as labels. We do not impose a strict constraint on the exact count $K$ sicne the entity-level personalization scenario implies that the frequency of entity occurrences may vary dramatically from one document to the other, which makes it difficult to set a strict limit. For instance, an entity type may occur more frequent in some documents and less so in others. The right area of Figure 1 shows an example $N$-way soft-$K$-shot VDER task. The **goal of task** $\mathcal{T}$ is to obtain a model that assigns each token as either

one of $\mathcal{E}$ (task-personalized entity types) or 0 (background or out-of-task entity types), based on the few labeled entity occurrences for those in $\mathcal{E}$ in support set $S$, such that the model achieves high performance on the query set $Q$.

**Distribution over FVDER Tasks.** Based on the above formulation for a single FVDER task, we further formulate a task distribution $P(\mathcal{T})$ over FVDER tasks. Assume there is a large pool of entity types $\mathcal{C}$ corresponding to the domain of $P(\mathcal{T})$. For any task $\mathcal{T}_i = \{S_i, Q_i, \mathcal{E}_i\} \sim P(\mathcal{T})$, its target entity types come from the class pool $\mathcal{E}_i \subset \mathcal{C}$.

**Global Objective.** The *global objective* is to train a meta-learner for $P(\mathcal{T})$ such that any task $\mathcal{T}_i \sim P(\mathcal{T})$ can take advantage of it and then quickly obtain a good *task-personalized* model. Following (Finn et al., 2017; Chen et al., 2021), to train the meta-learner, we simulate a meta-level dataset consisting of example FVDER task instances from $P(\mathcal{T})$. Figure 1 shows an overview of the dataset simulation of $P(\mathcal{T})$. Specifically, a meta-learner is trained from the experiences of solving a set of meta-training tasks $\mathcal{D}_{meta}^{trn} = \{\mathcal{T}_1, \mathcal{T}_2 ... \mathcal{T}_{\tau_{trn}}\}$ over a set of base classes $\mathcal{C}_{base} \subset \mathcal{C}$, where each training task is from the base classes $\mathcal{E}_i \subset \mathcal{C}_{base}$. The experiences are given in the form of the ground truth labels of query sets. That is, the query sets of training tasks are treated as validation sets, $Q_i = \{(X_j^*, Y_j^*)\}_{j=1}^{M_{qi}}$ for $\forall \mathcal{T}_i \in \mathcal{D}_{meta}^{trn}$. To evaluate the performance of the meta-learner on solving FVDER tasks with *novel* entity types $\mathcal{C}_{novel} = C \setminus \mathcal{C}_{base}$, we will *individually* train a set of meta-testing tasks $\mathcal{D}_{meta}^{test} = \{\mathcal{T}_1^*, \mathcal{T}_2^* ..., \mathcal{T}_{\tau_{tst}}^*\}$,

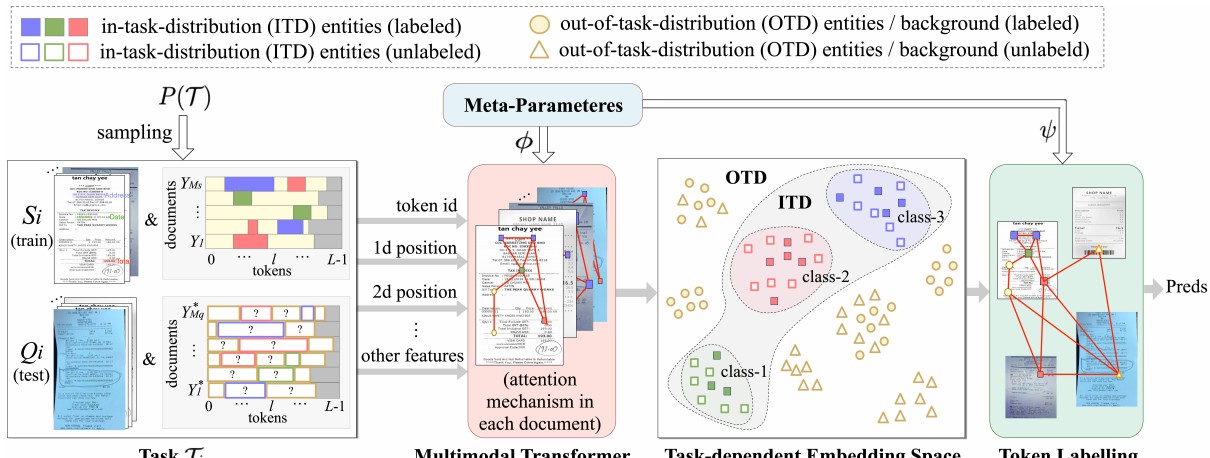

Figure 2: The proposed task-aware meta-learning framework. The framework is applicable to both the metric-based method (aiming to learn $\phi$) and gradient-based method (aiming to learn $\{\phi, \psi\}$).

where each testing task $\mathcal{E}_i^* \subset \mathcal{C}_{novel}$. The query sets of meta-testing tasks are unlabelled testing data, that is, $Q_i^* = \{X_j^*\}_{j=1}^{M_{qi}^*}, \forall \mathcal{T}_i^* \in \mathcal{D}_{meta}^{test}$.

## 3 Methodology

We propose a meta-learning (i.e., learning-to-learn) framework to solve the entity-level few-shot VDER tasks. Different from the recent advancements based on pre-training or prompts (Wang and Shang, 2022; Wang et al., 2023b), meta-learning helps to significantly promote *quick* adaptation and improve model *personalization* on task-specific entity types.

The proposed framework consists of three components: **(1)** a multimodal *encoder* (Section 3.1) that encodes the document images within a task into a *task-dependent embedding space* (Section 3.2); **(2)** a *token labelling* function (Section 3.3); and **(3)** a *meta-learner* built upon the encoder-decoder model, where we propose two task-personalized meta-learning methods (Section 3.4). Figure 2 shows an overview of the framework.

### 3.1 Multimodal Encoder

We consider an encoder network represented by a parameterized function $f_\phi^{enc}$ with parameters $\phi$. The encoder aims to capture the cross-modal semantic relationships between tokens in a document image. To achieve this, we employ a BERT-base Transformer (Kenton and Toutanova, 2019) with an additional positional embedding layer for the 2d position of each input token, through which the complex spatial structure of the input document can be incorporated and then interacted with the textual contents via attention mechanisms. The embedding of token $l$ in the document image $j$ of task $\mathcal{T}_i$

is computed as $\mathbf{h}_{ijl} = f_\phi^{enc}(\mathbf{x}_{ijl}|X_{ij})$. In practice, before meta-training, the multimodal Transformer is pretrained on the IIT-CDIP dataset (Harley et al., 2015). Details can be found in Appendix C.1.

### 3.2 Task-dependent Embedding Space

Through the multimodal encoder, each task $\mathcal{T}_i$ is encoded to a *task-dependent embedding space*. As illustrated in Figure 2, on the task-dependent embedding space, there are all the token embeddings in the task: $H_i = \{\mathbf{h}_{ijl}|l \in [L], (X_j, Y_j) \in S_i \cup Q_i\}$.

There are several properties on the task's embedding space: **(1)** *First*, in addition to *in-task distribution* (ITD) entities from the target classes, there exists a large portion (nearly 90% as observed in our dataset FewVEX) of *out-of-task distribution* (OTD) entities or background, which serve as the context for ITD entities but dominate the task's embedding space. **(2)** *Second*, the OTD entities follows a *multi-mode* distribution $P_i^{\text{OTD}}$ that consists of several unimodal distributions, each of which represents an outlier entity type aside from ITD. **(3)** *Finally*, it is not guaranteed that each unimodal component of $P_i^{\text{OTD}}$ is observable in the train set $S_i$–in many cases, an OTD entity type could occur in the query documents but is absent in the support documents. To sum up, the OTD distribution in a $N$-way $K$-shot FVDER task is complex, dominates the entire task, and may vary between documents.

### 3.3 Token Labelling

On the basis of the task-dependent embedding space, the token labelling or decoding process can either leverage a *parameterized* decoder $f_\psi^{dec}$ that acts as the classification head, or rely on *non-*

*parametric* methods, like nearest neighbors.

### 3.4 Task-aware Meta Learners

We consider two main categories of the meta-learning approaches: the gradient-based and the metric-based meta-learning, on each of which we propose our own methods. We specifically pay attention to two properties when solving the entity-level $N$-way $K$-shot FVDER tasks: **1) Few-shot out-of-task distribution detection**, which aims to distinguish the ITD (i.e., the target $N$ entity types) against the OTD (i.e., background or any outlier entity type). **2) Few-shot token labelling for in-task distribution tokens**, which assigns each ITD token to one of the $N$ in-task entity types.

#### 3.4.1 Task-aware ContrastProtoNet

We first focus on *metric-based* meta-learning (Snell et al., 2017; Oreshkin et al., 2018). The goal is to learn *meta-parameters* $\phi$ for the encoder network, generally shared by all tasks $\mathcal{T}_i \sim P(\mathcal{T})$, such that, on each task's specific embedding space, the distances between token points in $S_i$ and $Q_i$ are measured by some metrics, e.g., Euclidean distances.

**ProtoNet with or without Estimated OTD.** One of the most popular and effective metric-based meta-learning methods is the Prototypical Network (ProtoNet) (Snell et al., 2017). For each FVDER task $\mathcal{T}_i = \{S_i, Q_i, \mathcal{E}_i\}$, the prototype for each entity type $e \in \mathcal{E}_i$ can be computed as the mean embedding of the tokens from $S_i$ belonging to that entity type, that is, $\boldsymbol{\mu}_{i,e} = 1/|I_e^{\text{trn}}| \sum_{(j,l) \in I_e^{\text{trn}}} \mathbf{h}_{ijl}$, where $I_e^{\text{trn}}$ is a collection of the token indices for the type-$e$ tokens in the support set. For the out-of-task distribution (OTD), one may consider to estimate its mean embedding as an extra 0-type prototype: $\overline{\boldsymbol{\mu}}_i = 1/|I_{\text{OTD}}^{\text{trn}}| \sum_{(j,l) \in I_{\text{OTD}}^{\text{trn}}} \mathbf{h}_{ijl}$.

**Challenges.** A problem of the vanilla methods is that there is no specific mechanism distinguishing the ITD entities against the OTD entities, which are weakly-supervised and partially observed from a multi-mode distribution $P_i^{\text{OTD}}$. The prototype $\overline{\boldsymbol{\mu}}_i$ is a biased estimation of the mean of $P_i^{\text{OTD}}$ and the covariance of $P_i^{\text{OTD}}$ can be larger than any of the ITD classes. In consequence, the task-specific ITD classes may not be clearly distinguished from the OTD classes on the task-dependent embedding space and most of tokens will be misclassified.

Regarding the above challenges, we propose a task-aware method that adopts two techniques to boost the performance.

**Meta Contrastive Loss.** During meta-training, we encourage the $N$ ITD entity types to be distinguished from each other as well as far away from any unimodal component of OTD. To achieve this, we adopt the idea from supervised contrastive learning (Khosla et al., 2020) to compute a *meta contrastive loss* (MCON) from each task, which will be further used to compute meta-gradients for updating the meta-parameters $\phi$. Intuitively, our meta-objective is that the query tokens from the ITD type-$e$ should be pushed away from any OTD tokens and other types of ITD tokens within the same task, and should be pulled towards the prototype $\boldsymbol{\mu}_{i,e}$ of support tokens and the other query tokens belonging to the same entity type. Formally, let $I_{\text{ITD}}^{\text{val}} = \{(j,l)|l \in [L], (X_j^*, Y_j^*) \in Q_i, y_{ijl}^* \in \mathcal{E}_i\}$ denote a collection of ITD validation tokens. The meta contrastive loss computed from $\mathcal{T}_i$ is

$$\mathcal{L}_i^{\text{MCON}} = \sum_{(j,l) \in I_{\text{ITD}}^{\text{val}}} \frac{-1}{|A^+(j,l)|} \sum_{\mathbf{v} \in A^+(j,l)} a_{ijl}(\mathbf{v})$$

$$a_{ijl}(\mathbf{v}) = \log \frac{\exp(\mathbf{h}_{ijl}^\top \mathbf{v})}{\sum_{\mathbf{u} \in A(j,l)} \exp(\mathbf{h}_{ijl}^\top \mathbf{u})}. \tag{2}$$

For each *anchor*, i.e., the ITD validation token $l$ in document $j$, we let $A^+(j,l) = \{\mathbf{h}_{irm}|(r,m) \in I_{\text{ITD}}^{\text{val}} \setminus \{(j,l)\}, y_{ijl}^* = y_{irm}^*\} \cup \{\boldsymbol{\mu}_{i,e}|e \in \mathcal{E}_i, y_{ijl}^* = e\}$ denote a collection of the *positive* embeddings/prototype for the anchor and let $A(j,l) = \{\mathbf{h}_{irm}|(r,m) \in I_{\text{ALL}} \setminus \{(j,l)\}\} \cup \{\boldsymbol{\mu}_{i,e}\}_{e \in \mathcal{E}_i}$ contain all the ITD/OTD embeddings and prototypes ($I_{\text{ALL}} = \{(j,l)|l \in [L], (X_j, Y_j) \in S_i \cup Q_i\}$) in $\mathcal{T}_i$.

**Unsupervised OTD Detector.** During the testing time for novel entity types, we adopt the non-parametric token-level nearest neighbor classifier, which assigns $\mathbf{x}_{ijl}$ the same label as the support token that is nearest in the task's embedding space:

$$\hat{y}_{ijl}^{\text{nn}} = \text{argmax}_{y_{irm} \text{ where } (r,m) \in I_{\text{ALL}}^{\text{trn}}} \mathbf{h}_{ijl}^\top \mathbf{h}_{irm}, \quad (3)$$

where $I_{\text{ALL}}^{\text{trn}} = \{(r,m)|m \in [L], (X_r, Y_r) \in S_i\}$. The ITD or OTD entity tokens in $Q_i$ should be closer to the corresponding ITD or OTD tokens in $S_i$ that belong to the same entity type. However, since the embedding space dependent on the support set is not sufficiently rich, the network may be blind to properties of the out-of-task distribution $P_i^{\text{OTD}}$ that turn out to be necessary for accurate entity retrieval. To tackle this, we exploit an unsupervised out-of-distribution detector (Ren et al., 2021) operating on the task-dependent embedding space,

in assistance with the classifier. Specifically, we define an OTD detector: $\hat{y}_{ijl} = 0$ if $r(\mathbf{h}_{ijl}) \geq R_i$; otherwise, $\hat{y}_{ijl} = \hat{y}_{ijl}^{nn}$, where $R_i$ is the task-dependent uncertainty threshold and $r(\mathbf{h}_{ijl})$ is defined as the OTD score of each token computed as its minimum Mahalanobis distance among the $N$ ITD classes: $r(\mathbf{h}_{ijl}) = \min_{e \in \mathcal{E}_i}(\mathbf{h}_{ijl} - \boldsymbol{\mu}_{i,e})^\top \Omega_{i,e}^{-1}(\mathbf{h}_{ijl} - \boldsymbol{\mu}_{i,e})$. Here, $\Omega_{i,e} = \sum_{(j,l) \in I_e^{\mathrm{trn}}}(\mathbf{h}_{ijl} - \boldsymbol{\mu}_{i,e})^\top(\mathbf{h}_{ijl} - \boldsymbol{\mu}_{i,e})$ is the covariance matrix for entity type $e$ computed from the type-$e$ tokens in the support set ($I_e^{\mathrm{trn}}$). The higher OTD score indicates the more likely the token belongs to the background.

### 3.4.2 Computation-efficient Gradient-based Meta-learning with OTD Detection

For *gradient-based meta learning*, the goal is to learn the meta-parameters $\theta = \{\phi, \psi\}$ globally shared over the task distribution $P(\mathcal{T})$, which can be fast fine-tuned for any given individual task $\mathcal{T}_i$.

**Computation-efficient Meta Optimization.** Although MAML (Finn et al., 2017) is the most widely adopted approach, the fact that it needs to differentiate through the fine-tuning optimization process makes it a bad candidate for Transformer-based encoder-decoder model, where we need to save a large number of high-order gradients for the encoder. Instead, we consider two alternatives which require less computing resources and more efficient. **ANIL** (Raghu et al., 2019) employs the same bilevel optimization framework as MAML but the encoder is not fine-tuned during the inner loop. The features from the encoder are reused in different tasks, to enable the rapid fine tuning of the decoder. **Reptile** (Nichol et al., 2018) is a first-order gradient based approach that avoids the high-order meta-gradients. To further boost training efficiency, we exploit Federated Learning (Tian et al., 2022; Chen and Zhang, 2022a) for meta-optimization of Transformer.

**Task-aware Hierarchical Classifier (HC).** A vanilla classifier can achieve high performance in the label-sufficient VDER. However, it turns out to be not robust in few-shot FVDER tasks because of the existence of the complicated out-of-task entities–the models usually either get overconfident on the $N$ IID entity types or fail to distinguish target entities from the OTD background. For this reason, we incorporate OTD detection into the decoder and propose a hierarchical classifier, which has two classifiers $\psi = \{\psi_1, \psi_2\}$: 1) *binary* classifier $f_{\psi_1}^{bin}$, so that all ITD tokens are classified

against OTD ones, and 2) *entity* classifier $f_{\psi_2}^{ent}$, so that ITD tokens are classified to one of the $N$ entity types of the task. Specifically, suppose $P_i^{\mathtt{OTD}}$ and $P_i^{\mathtt{ITD}}$ denotes the OTD and ITD of the task $\mathcal{T}_i$, respectively. The probability that the token $\mathbf{h}_{ijl}$ is from OTD is denoted as $P(y_{ijl} = 0) = f_{\psi_{i1}'}^{ent}(\mathbf{h}_l)$, which is used as the OTD score to weight the entity prediction. The probability that the token is the entity type-$e$ is computed as $P(y_{ijl} = e | \mathbf{x}_{ijl} \in P_i^{\mathtt{ITD}}) = (1 - P(y_{ijl} = 0))f_{\psi_{i2}'}^{ent}(\mathbf{h}_{ijl})_e$.

## 4 FewVEX Dataset

There is no existing benchmark specifically designed for task-personalized Entity-level $N$-way Soft-$K$-shot VDER. To facilitate future research on this problem, we create a new dataset, *FewVEX*.

**Source Collection[1]:** FewVEX is built from two source datasets: **FUNSD** (Jaume et al., 2019) contains images of forms annotated by the bounding boxes of 3 types of entities; **CORD** (Park et al., 2019) contains scanned receipts annotated by 6 superclasses which are divided into 30 fine-grained subclasses. From them, we collect 1199 document images ($\mathcal{D}$) annotated by 26 entity types ($\mathcal{C}$).

**Meta-learning Tasks:** We use $\mathcal{D}$ and $\mathcal{C}$ to construct FewVEX, represented by $\mathcal{D}_{meta} = \{\mathcal{D}_{meta}^{trn}, \mathcal{D}_{meta}^{tst}\}$ such that the testing tasks $\mathcal{D}_{meta}^{tst}$ focus on novel classes that are unseen in $\mathcal{D}_{meta}^{trn}$ during meta-training. To create this, we split $\mathcal{C}$ into two separate sets $\mathcal{C} = \mathcal{C}_{base} \cup \mathcal{C}_{novel}, \mathcal{C}_{base} \cap \mathcal{C}_{novel} = \emptyset$, where $\mathcal{C}_{base}$ is used for meta-training and $\mathcal{C}_{novel}$ for meta-testing.

**Single Task Generation:** Following the definition in Eq.(1), each individual entity-level $N$-way soft-$K$-shot VDER task $\mathcal{T} = \{S, Q, \mathcal{E}\}$ in either $\mathcal{D}_{meta}^{trn}$ or $\mathcal{D}_{meta}^{tst}$ can be generated through the following steps. **(1)** *Task-personalized Class sampling.* The task's target classes $\mathcal{E}$ are generated by randomly sampling $N$ entity types from either $\mathcal{C}_{base}$ (for the training task) or $\mathcal{C}_{novel}$ (for the testing task). **(2)** *Document sampling.* Given the $N$ target classes, we then collect document images that satisfies the $N$-way, soft $K$-shot entity occurrences (as in Appendix 1). **(3)** *Annotation Conversion.* A task only focuses on its specific $N$ rarely-present entity types. The entities in the original annotated documents, whose class do not belong to $\mathcal{E}$, are replaced with the background 0 class[2].

---

[1]Due to page limit, details are moved to Appendix B.1.
[2]Due to page limit, details can be found in Appendix B.2.2.

| Datasets | Meta Training (from $\mathcal{C}_{base}$) | | | Meta Testing (from $\mathcal{C}_{novel}$) | | | Range of $N$ |
|---|---|---|---|---|---|---|---|
| | Domains | # Entity Types | # Tasks | Domains | # Entity Types | # Tasks | |
| FewVEX(S) | CORD | 18 | 3000 | CORD | 5 | 128 | [1, 5] |
| FewVEX(M) | CORD+FUNSD | 20 | 3000 | CORD+FUNSD | 6 | 256 | [1, 6] |

Table 2: Statistics of two variants of FewVEX. From each dataset, we can test different $N$-way $K$-shot settings.

**Proposed Datasets:** We construct two variants of FewVEX. **FewVEX(S)** focuses on single-domain receipt understanding, where $\mathcal{C}_{base}$ and $\mathcal{C}_{novel}$ are split from the 23 entity types in CORD. **FewVEX(M)** focuses on a combination of receipt and form domains, where $\mathcal{C}_{base}$ contains 18 classes from CORD and 2 from FUNSD; $\mathcal{C}_{novel}$ contains the other 5 classes in CORD and 1 in FUNSD. The statistics of FewVEX is summarized in Table 2.

**Future Extension:** While CORD and FUNSD currently serve as the source datasets for FewVEX, we anticipate that future enhancements such as expanding the number of entity types ($|\mathcal{C}|$) and diversifying documents ($|\mathcal{D}|$) will lead to a better version of FewVEX. We introduce **Cross-document Rejection (XDR)** sampling to facilitate this improvement. XDR samples the train/test documents of each task in a way such that cooperatively ensures a specific range of entity occurrences per class, which mimics real-world user annotation behaviors motivated by class-balanced requirements. In the future, with access to open-source VDER datasets containing a wide array of classes and documents, XDR will enable the automated generation of numerous distinct task simulations. The pseudocode of XDR is shown in Algorithm 1 in the appendix.

## 5 Experiments

**Setups:** We compare the proposed framework with aforementioned meta-learning baselines on FewVEX. Data generation and methods are implemented using JAX and Tensorflow. All experiments ran on 32 TPU devices. We use the Adam optimizer to update the meta-parameters. For gradient based methods, we use vanilla SGD for the inner-loop optimization and fix 15 SGD updates with a constant learning rate of 0.015. Setup details and hyperparameters are available in Appendix C.4.

**Evaluation Metrics:** We consider two types of quantitative metrics. **(1) Overall Performance**: following (Xu et al., 2020), we use the precision (**P**), recall (**R**), and micro **F1**-score over meta-testing tasks to measure the accuracy of entity retrieval. **(2) Task Specificity (TS)**: to evaluate how

well the trained meta-learners can distinguish in-task distribution (ITD) from out-of-task distribution (OTD) for any novel given task, we plot ROC curves and calculate **AUROC** (Xiao et al., 2020) using the ITD scores over meta-testing tasks. A random guessing detector outputs an AUROC of 0.5. A higher AUROC indicates better TS performance.

### 5.1 Main Results

Table 3 reports the results on FewVEX(S). Under the same $N$ and $K$ setups, traditional meta-learning methods fail to balance the precision and recall performances: ANIL and Reptile using vanilla decoders achieved high precision but tended to perform low recall; the vanilla Prototypical Networks tended to be opposite: low precision but high recall. In contrast, ANIL+HC, Reptile+HC and Contrast-ProtoNet, achieved better precision-recall balance and thus higher F1 scores and TS, proving that detecting and alleviating the influence of out-of-task distribution can improve task personalization and accuracy. Such phenomenon is also illustrated in Figure 3 and Figure 5 in Appendix D.2, where we plot ROC curves and tSNE visualizations of token embeddings after task adaptation. Comparing our methods against baselines, we observe an elevation in the curves and more distinct boundaries between OTD and ITD and between ITD classes.

The reasons are as follows. *First*, ANIL and Reptile treat the dominant OTD instances as an extra class as well. The problem turns out the imbalanced classification in meta-learning, one of the challenges in few-shot VDER tasks. By using an OTD detector, ANIL+HC and Reptile+HC can faster adapt to the task-specific boundary between OTD and ITD. Overall, this potentially increase the recall and task specificity score and the overall F1 score. *Second*, for the vanilla metric-based methods, where OTD instances are treated as one extra class, the ITD testing instances tend to be close to ITD class centers so that we have high recall. However, OTD instances dominate the task. It is possible that some OTD testing instances are closer to ITD centers than the OTD class center (the average center of multiple OTD classes) so that most

| Methods | 4-way 1-shot | | | | 4-way 4-shot | | | | 5-way 2-shot | | | |
| --- | --- | --- | --- | --- | --- | --- | --- | --- | --- | --- | --- | --- |
| | Overall | | | TS | Overall | | | TS | Overall | | | TS |
| | **P** | **R** | **F1** | **AUROC** | **P** | **R** | **F1** | **AUROC** | **P** | **R** | **F1** | **AUROC** |
| ProtoNet | 0.02 | 0.10 | 0.03 | N/A | 0.02 | 0.09 | 0.03 | N/A | 0.02 | 0.09 | 0.03 | N/A |
| ProtoNet+EOD | 0.13 | 0.47 | 0.21 | N/A | 0.11 | 0.58 | 0.23 | N/A | 0.11 | 0.35 | 0.17 | N/A |
| **ContrastProtoNet** | 0.54 | 0.43 | **0.47** | **0.59** | 0.61 | 0.59 | **0.60** | **0.89** | 0.49 | 0.41 | **0.44** | **0.62** |
| Reptile | 0.48 | 0.10 | 0.15 | 0.58 | 0.62 | 0.44 | 0.51 | 0.67 | 0.39 | 0.09 | 0.14 | 0.59 |
| ANIL | 0.39 | 0.19 | 0.25 | 0.56 | 0.54 | 0.44 | 0.50 | 0.87 | 0.35 | 0.13 | 0.19 | 0.61 |
| **Reptile+HC** | 0.35 | 0.13 | 0.20 | 0.63 | 0.63 | 0.65 | **0.64** | **0.98** | 0.34 | 0.12 | 0.18 | 0.65 |
| **ANIL+HC** | 0.40 | 0.58 | **0.50** | **0.95** | 0.47 | 0.59 | 0.51 | 0.98 | 0.38 | 0.56 | **0.46** | **0.92** |

Table 3: Performance on 4-way 1-shot, 4-way 4-shot, and 5-way 2-shot settings of FewVEX(S).

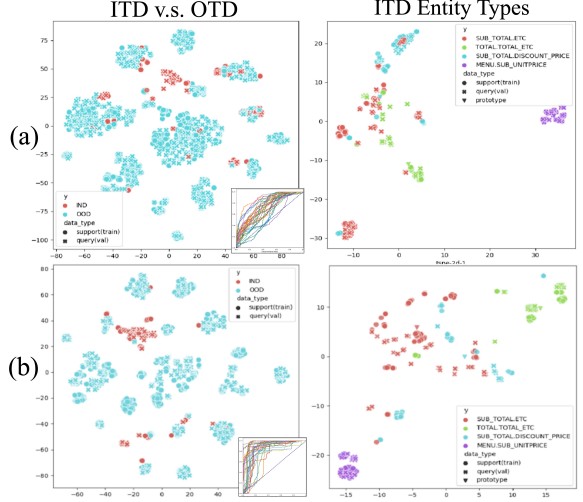

Figure 3: tSNE visualization of the learned embedding space for a randomly-selected meta-testing task, comparing (a) vanilla ProtoNet and (b) ContrastProtoNet methods, under the 4-way 4-shot setting of FewVEX(S).

of them are misclassified as one of ITD classes, i.e., low precision. In opposite, ContrastProtoNet does not make any assumption on the OTD distribution; instead, we enforce OTD to be far away from ITD classes and classify via token-level similarities while considering probabilistic uncertainty.

## 5.2 Class Structure Disentanglement

We examine the explanability and disentanglement of the learned representations (generated by the meta-parameters of encoder). Figure 3 shows tSNE visualizations of the learned embedding space of a selected task. Overall, by comparing Figure 3 to Table 3, the higher performance appears to be consistent with more disentangled clusters. Moreover, from the first column containing ITD (*red*) tokens and OTD (*blue*) tokens, we observe that the blue points dominate the embedding space and comprises multiple clusters, which demonstrates the out-of-task distribution is multimodal, making

it hard to identify in-task entities. Further, we try to understand the disentangled structure of classes from the clusters. In the right column in Figure 3, we zoom into the four ITD classes, where purple, red, blue, and green points denotes the task-specific four entity types, respectively. We observe that "menu (sub_uniprice)" (violet) is far away from the other three classes, while the other three classes are slightly entangled. Such class structure represents the relationships between these entity types, which is explainable: the red and blue classes belong to the same superclass sub_total; the green and red are both etc information.

## 5.3 Multi-domain Few-shot VDER

Table 4 reports the 4-way 2-shot results on the mixed-domain FewVEX(M), which combines receipts with forms for few-shot learning. The results slightly underperform those under the single-domain setting. A reason could be that the structure of forms is different from that of receipts and it is challenging to find the good meta-parameters for both domains. Moreover, the number of classes in the form domain is much smaller than that in the receipt domain. Such imbalanced class combination would push the meta-parameters to adapt to the relative prominent domain.

| Methods | P | R | F1 | AUROC |
| --- | --- | --- | --- | --- |
| ProtoNet | 0.02 | 0.10 | 0.03 | N/A |
| ProtoNet+EOD | 0.18 | 0.46 | 0.26 | N/A |
| **ContrastProtoNet** | 0.54 | 0.46 | **0.50** | **0.85** |
| Reptile | 0.45 | 0.17 | 0.25 | 0.57 |
| ANIL | 0.39 | 0.19 | 0.26 | 0.56 |
| **Reptile+HC** | 0.42 | 0.23 | 0.30 | 0.88 |
| **ANIL+HC** | 0.44 | 0.56 | **0.49** | **0.97** |

Table 4: Performance on 4-way 2-shot FewVEX(M).

## 6 Related Works

Research related to Visually-rich Documents (VD) have emerged as significant topics in NLP. Here,

we briefly review the prior research of **(1)** models for *general* VD understanding; **(2)** the *particular* Entity Retrieval (ER) task for VD and existing Few-shot VDER methods; **(3)** the methodology-level related works in general few-shot learning[3].

**General VD Understanding.** Pretrained LLMs for *general* VD understanding have shown strong performance in general understanding of visually-rich multimodal documents, and therefore, can serve as *pretrained prior* for Few-shot VDER. There are many LLM candidates our framework can use as the pretrained encoder, such as LayoutLM (Xu et al., 2020), which extends the standard BERT (Kenton and Toutanova, 2019), and the recent LayoutLMv3 (Huang et al., 2022) and DocGraphLM (Wang et al., 2023a), which show improvements by using advanced cross-modal alignment or local-global position embeddings. In this paper, we use the basic BERT model for experiments since our focus is how to improve the *post* fine-tuning on few-shot downstream tasks, without a restrict on the specification of LLM type. Extending this research to other pretrained Document Understanding LLMs could be one of future works.

**Few-shot VD Entity Retrieval.** The *particular* Entity Retrieval (ER) tasks for VD have been studied for many years using Deep Neural Networls, Graph Neural Networks, or traditional models (Zhang et al., 2020; Shi et al., 2023), or empowered by the contextual prior knowledge provided by VD-understanding LLMs (Xu et al., 2021; Lee et al., 2022; Hong et al., 2022). VDER in the few-shot scenarios pose unique challenges such as achieving task personalization with limited annotation, yet has garnered comparatively less attention in prior research. Recent advancements in Few-shot VDER predominantly rely on pretrained LLMs and prompt design, followed by fine tuning on a small number of VD documents (Wang et al., 2021b; Wang and Shang, 2022). Despite their success, this paper explores a complementary research perspective. While previous works address the situation where the entity label space is fixed over tasks and entity occurrences do not shift a lot, we tackle a different application situation–every few-shot task is user-specific, focusing on a small subspace of interested entity types (entity-level task personalization), and entity occurrences vary significantly between tasks and documents.

---

[3]An extended version of Related Works is in Appendix A.

**General Few-shot Learning.** Few-shot Learning (FSL) has been studied in various AI/ML domains (Song et al., 2023). In CV or NLP domains, there are two FSL tasks closely related to Few-shot VDER: **(1)** Few-shot object detection or segmentation (Köhler et al., 2023; Antonelli et al., 2022) aims at localizing objects in visual data, where each object can be treated as an entity in VDER; and, **(2)** Few-shot Named Entity Recognition (NER) aims at labelling tokens within a contextual text sequence (Li et al., 2022; Huang et al., 2021). While few-shot NER and object detection algorithms can provide inspirations for few-shot VDER, the challenges we face and methodology details are relatively different. Beyond them, Multimodal Few-shot Learning (*MFSL*) utilizes complementary information from multiple modalities to improve a unimodal FSL (Chen and Zhang, 2021; Lin et al., 2023). The scope of this paper falls within the field of MFSL. While existing FSL/MFSL approaches can be categorized into *meta*-learning approaches (Snell et al., 2017; Finn et al., 2017) and *non-meta* LLM pretraining-and-fine-tuning approaches (Brown et al., 2020), we employ the benefits from both LLM prior knowledge and the meta-learning for task-personalized fine-tuning. Furthermore, to enhance task specificity performance of few-shot VDER, we employ Few-shot *Out-of-distribution (OOD) Detection*, which itself is a recently emerged task (Le et al., 2021).

# 7 Conclusions

In this paper, we studied the multimodal few-shot learning problem for VDER. We started by proposing a new formulation of the FVDER problem to be an entity-level, $N$-way soft-$K$-shot learning under the framework of meta learning as well as a new dataset, FewVEX, which is designed to reflect the practical problems. To solve the new task, we exploited both metric-based and gradient-based meta-learning paradigms, along with a new technique we proposed to enhance task personalization via out-of-task-distribution awareness. The experiments showed that the proposed methods achieve major improvements over the baselines for FVDER.

For future works, our approaches might be improved in the following directions: (1) A more robust algorithm that distinguishes between the OTD and ITD. (2) An advanced decoding process considering graphical structures or implicit correlations between entity instance within each task. (3) Exploring the causal role of pretrained models.

## Acknowledgements

We would like to express sincere appreciation to all those who contributed to this research. Special thanks to all the reviewers for their constructive feedback and comments, which greatly improved the quality of this paper.

## Limitations

There exists a few limitations to this work. Firstly, the derived dataset is based on the current open source ones for document understanding, which are small in their size and has very limited amount of classes. A dedicated dataset that is built specifically for the purpose of studying few-shot learning for document entity retrieval is needed. Secondly, the scope of our current studies is limited to non-overlapping entities. The performance of the models under nested and entities with overlapping ground truth is yet to be examined.

## Ethics Statements

The dataset created in this paper was derived from public datasets (i.e., FUNSD, CORD) which are publicly available for academic research. No data collection was made during the process of making this work. The FUNSD and CORD datasets themselves are a collection of receipts and forms collected and released by a third party paper which has been widely used in the field of visually rich document entity retrieval research and is not expected to contain any ethnics issues to the best of our knowledge.

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

# Appendix

## A    Related Works

We review related works corresponding to the Few-shot VDER tasks in Section A.1 and then review methodology-level related works in Section A.2.

### A.1    Visually-rich Document Related Works

**V**isually-rich **D**ocuments (VD) are a vital category of *multimodal* data in the field of **Document AI**, typically consisting of texts, images, and layout structure of contents. Research and industrial applications pertaining to VD have emerged as significant topics in NLP over the past decade. Here we review the prior research works for **VD-related tasks**, including **(1)** LLMs for *general* VD understanding; **(2)** the *particular* Entity Retrieval (ER) task in VD and its prior work; and **(3)** existing works considering VDER in *few-shot scenarios*.

**Visually-rich Document Understanding LLMs.** Large Language Models (LLMs) have shown strong performance in general understanding of visually-rich multimodal documents, and therefore, can serve as *pretrained prior* for Few-shot VDER. For this reason, here we review several LLM candidates we can use. LLMs for *text-image-layout* document understanding have emerged since LayoutLM (Xu et al., 2020), which extends the standard BERT (Kenton and Toutanova, 2019) by additional layout information obtained from OCR (Chaudhuri et al., 2017) preprocessing. After this, SelfDoc (Li et al., 2021), UDoc (Gu et al., 2021), LayoutLMv2 (Xu et al., 2021), TILT (Powalski et al., 2021), DocFormer (Appalaraju et al., 2021), LiLT (Wang et al., 2022a), and LayoutLMv3 (Huang et al., 2022), show improvements by using cross-modal alignment or modern feature encoders for the image modality (e.g., ResNets (He et al., 2016) and dVAE (Ramesh et al., 2021)). Very recent works, UniFormer (Yu et al., 2023), LayoutMask (Tu et al., 2023), and DocGraphLM (Wang et al., 2023a), employ token-level strategies like local text-image alignment, local position embeddings, and graph representation, to improve the modeling. In this paper, we use the basic BERT model for experiments since our focus is how to improve the *post* fine-tuning on few-shot downstream tasks, without a restrict on the specification of LLM type. Extending this research to other pretrained Document Understanding LLMs could be one of future works.

**Entity Retrieval in Visually-rich Documents.** Visually-rich Document Entity Retrieval (VDER) aims at detecting bounding boxes for specific types of key information within scanned or digitally-born documents, which has garnered significant attention from researchers. While there are technical differences between Data-sufficient VDER and Few-shot VDER, the former offers foundational solutions and serves as a valuable baseline framework. Therefore, reviewing existing Data-sufficient VDER techniques is worthwhile. At early years, Deep Neural Networks (e.g., RNNs, CNNs) have been widely employed in addressing VDER tasks (Huang et al., 2019; Zhang et al., 2020). Later, Graph Neural Networks (GNNs) (Liu et al., 2019; Gal et al., 2020; Carbonell et al., 2021; Shi et al., 2023) have gained substantial attention for their effectiveness in tackling the structural layout information. Recent works empower LLMs of general VD Understanding to incorporate additional contextual prior knowledge and fine tune on VDER tasks (Xu et al., 2021; Garncarek et al., 2021; Lee et al., 2022; Hong et al., 2022). Beyond these research, this paper focuses on the Few-shot VDER cases with limited annotation, which pose unique challenges in achieving *task personalization* with scarce data, addressing *shot imbalance*, and handling task complexity due to *out-of-task-distribution* entities.

**Few-shot VDER.**    There has been rare discussions on VDER in few-shot scenarios. Recent works primarily focus on pretraining LLMs and prompt design such that they can *then* be fine tuned on a small number of VD documents (Wang et al., 2021b; Wang and Shang, 2022; Xu et al., 2021; Huang et al., 2022). Despite their success, our paper explores a complementary research perspective: **(1)** While previous works emphasize first-stage LLM pretraining, our work focuses on the second-stage few-shot adaptation algorithm. **(2)** We tackle a different application *situation* from the previous works. Previous works address a specific few-shot situation where the entity label space is fixed and entity occurrences do not vary a lot from one document to another. In contrast, our research tackles a different situation, where the entity label spaces and entity occurrences vary significantly between tasks and documents, enabling entity-level task personalization (i.e., each personal few-shot task is only interested in a small subset of entity types). Both situations can happen in the real world. This paper addresses the second unexplored one.

## A.2 General Few-shot Learning

Next, we review the methodology-level related works from the other domains that are closely related to, but beyond, VDER. First, we briefly review of general Multimodal Few-shot Learning *algorithms*. Then, we review literature in the fields of CV and NLP that address **non-VDER** but closely related tasks, including: **(1)** *vision-only* Few-shot Object Detection and Segmentation in the CV domain; **(2)** *text-only* Few-shot Named Entity Recognition in the NLP domain; and **(3)** the *general* Few-shot Out-of-distribution Detection.

**Multimodal Few-shot Learning.** Few-shot Learning (FSL) has been studied in various AI/ML domains, such as CV, NLP, healthcare, etc (Song et al., 2023). Multimodal Few-shot Learning (MFSL) jointly utilizes complementary information from multiple modalities to improve a uni-modal task (Pahde et al., 2021; Chen and Zhang, 2021; Lin et al., 2023). The scope of this paper falls within the domain of MFSL, with a specific emphasis on multimodal documents. Existing MFSL work falls into two categories: *non-meta* learning methods and *meta*-learning approaches. The former typically involves a two-stage training process LLM pretraining and fine-tuning or prompt learning (Wang et al., 2023b). On the other hand, meta-learning approaches formulate a task-level distribution, and then, learn task-adaptive metric functions (Snell et al., 2017; Oreshkin et al., 2018; Koch et al., 2015; Vinyals et al., 2016) or employ bilevel optimization to learn meta-parameters for fast task-adaptive fine tuning (Finn et al., 2017; Yoon et al., 2018; Rusu et al., 2019; Chen and Zhang, 2022b). The proposed framework benefits from both LLMs and the meta-learning paradigm by being build upon the LLMs and then using meta-learning for task-adaptive fine-tuning.

**Few-shot Object Detection and Segmentation.** Few-shot object detection and Few-shot Segmentation are CV tasks that aim at recognizing and localizing novel objects or semantics in an image with only a few training examples (Wang et al., 2020; Köhler et al., 2023; Antonelli et al., 2022). The output of entity retrieval from document images consists of bounding boxes for entities, allowing it to be formulated as an object detection or segmentation problem in the CV domain, where each object is treated as an entity (Shen et al., 2021). However, while few-shot object detection and segmentation

algorithms (Sun et al., 2021) can provide inspirations for Few-shot VDER, there are still gaps between Few-shot VDER and these fields: the lack of Few-shot VDER datasets in the form of object detection or segmentation tasks; and, the scales of entity objects are often much smaller than the out-of-distribution background objects.

**Few-shot Sequence Labeling.** This paper adopts the few-shot sequence labeling paradigm as introduced by (Wang et al., 2021a). Many other NLP tasks have also embraced this paradigm, including Few-shot Named Entity Recognition (NER) (Li et al., 2022). Few-shot NER that focus on limited entity occurrences was initially introduced in Few-NERD (Huang et al., 2021), and later, (Ma et al., 2022) proposed a meta-learning approach to address this task. While Few-shot NER tasks are text-only, devoid of visual and layout modalities, and typically involve short texts, Few-shot VDER at the entity level presents a greater challenge, where the difficulty lies in effectively integrating layout structure and visual information and achieving task personalization from out-of-distribution background.

**Few-shot Out-of-Distribution Detection.** Machine learning models, when being deployed in open-world scenarios, have shown to *erroneously* produce high posterior probability for out-of-distribution (OOD) data. This gives rise to OOD detection that identifies unknown OOD inputs so that the algorithm can take safety precautions (Ming et al., 2022). Recently, motivated by real-world applications, OOD detection in the *few-shot* settings increasingly attracts attentions (Le et al., 2021; Jeong and Kim, 2020; Wang et al., 2022b), which faces new challenges such as a lack of training data required for distinguishing OOD from task-specific class distribution. In this paper, the proposed framework for Few-shot VDER employs few-shot OOD detection to improve performance: to prevent the prediction of background context as one of task-personalized entities, we encourage task-aware fine tuning to exclude statistically informative yet *spurious* features in the support set.

## B FewVEX Dataset

Since there is no dataset specifically designed for the Few-shot VDER task defined in Section 2, we construct a new dataset, FewVEX, to benchmark and evaluate Few-shot VDER tasks.

## B.1 Collection of Entity Types and Documents

First, we collect the entity types $\mathcal{C}$ associated with the task distribution $P(\mathcal{T})$ and a set of document images $\mathcal{D}$ annotated by these entity types.

We consider two source datasets that are widely used in normal large-scale document understanding tasks such as entity recognition, parsing, and information extraction. The first one is the Form Understanding in Noisy Scanned Documents (FUNDS) dataset (Jaume et al., 2019) comprises 199 real, fully annotated, scanned forms, with a total of three types of entities (i.e., questions, answers, heads). The second one is the Consolidated Receipt Dataset for post-OCR parsing (CORD) dataset (Park et al., 2019). CORD consists of 1000 receipt images of texts and contains 6 superclasses (menu, void menu, subtotal, void total, total, and etc) which are divided into 30 fine-grained subclasses. For different entity types, the total numbers of entity occurrences over the CORD images are highly imbalanced, ranging from 1 occurrence of entity "void menu (nm)" to 997 occurrences of "menu (price)".

From the two datasets, we obtain a combined source dataset denoted as $\mathcal{D}$, which contains 1199 unique document images with original annotations on 33 classes. However, we observe that some fine-grained classes in CORD occurs in less than $max_i(M_{si} + M_{qi})$ images, the maximum number of documents within individual tasks. This will result in a large amount of repetitive usage of the same documents within one task and between different tasks. Therefore, we further sort the 33 classes by the number of unique document images where they occur and then discard three entity types that occurs in low frequency.

To sum up, we finally have a total of $|\mathcal{C}| = 30$ entity types and $|\mathcal{D}| = 1199$ unique document images annotated by these entity types. The pie chart (on the left) in Figure 1 illustrates the number of occurrences of the final entity types.

## B.2 Collection of Training and Testing Tasks

We simulate a distribution of tasks $P(\mathcal{T})$ in FewVEx. We create a meta-learning dataset $\mathcal{D}_{meta} = \{\mathcal{D}_{meta}^{trn}, \mathcal{D}_{meta}^{tst}\}$, consisting of a meta-training set $\mathcal{D}_{meta}^{trn} = \{\mathcal{T}_1, \mathcal{T}_2...\mathcal{T}_{\tau_{trn}}\}$ containing $\tau_{trn}$ training tasks and a meta-testing set $\mathcal{D}_{meta}^{test} = \{\mathcal{T}_1^*, \mathcal{T}_2^*..., \mathcal{T}_{\tau_{tst}}^*\}$ containing $\tau_{tst}$ testing tasks. Each task instance follows the $N$-way $K$-shot FVDER task setting that pays attention to $N$ personalized entity types.

## B.2.1 Entity Type Split

To ensure that testing tasks in $\mathcal{D}_{meta}^{tst}$ focus on novel classes that are unseen during meta-training $\mathcal{D}_{meta}^{trn}$, we should split the total entity types $\mathcal{C}$ into two separate sets $\mathcal{C} = C_{base} \cup \mathcal{C}_{novel}, \mathcal{C}_{base} \cap \mathcal{C}_{novel} = \emptyset$ such that $\mathcal{C}_{base}$ is used for meta-training and $\mathcal{C}_{novel}$ for meta-testing.

Specifically, we use a split ratio $\gamma$ to control the number of novel classes and randomly choose $\gamma|C|$ entity types from $\mathcal{C}$ as $\mathcal{C}_{novel}$. Then, $\mathcal{C}_{base} = \mathcal{C} \setminus \mathcal{C}_{novel}$. Note that for the cases that some entity types occurs in less number of documents than the others, we set a threshold $U$ and any entity type that occurs in less than $U$ documents are forced to be one of the novel classes.

## B.2.2 Single N-way K-shot Task Simulation

Each individual task $\mathcal{T} = \{S, Q, \mathcal{E}\}$ in either $\mathcal{D}_{meta}^{trn}$ or $\mathcal{D}_{meta}^{tst}$ can be generated by the following steps (summarized in Algorithm 1).

**Personalized Class Sampling.** The target classes of task $\mathcal{E}$ is generated by randomly sampling $N$ entity types from either $\mathcal{C}_{base}$ (for the training task) or $\mathcal{C}_{novel}$ (for the testing task).

**Document Sampling.** Given the $N$ target classes, we then collect document images that satisfies the few-shot setting defined in Section 2. However, one problem of document sampling from the original corpus is the *inefficiency*. It is because, for each task, only a small number of documents that contain the corresponding classes can be the candidate documents of the task. For example, if each document contains only a small number of entity types, the majority of documents would be rejected. To improve sampling efficiency, one strategy is to count entities in each document in advance and, for each entity type, all the candidate documents that contain this type are temporally stored in a new dataset. We only look at the task-specific candidate datasets $\mathcal{D}^{\mathcal{E}} = \{\mathcal{D}^e | \forall e \in \mathcal{E}\}$, where $\mathcal{D}^e = \{(X, Y) | \forall (X, Y) \in \mathcal{D} \text{ if } e \in Y\}$. We proposed Cross-document Rejection sampling in Algorithm 1, which randomly sample $M_s$ documents such that the total number of entity instances is satisfied–that is, $K \sim \rho K$ shots per entity type. Likewise, we sample $M_q$ documents for $Q$, such that there are $K_q \sim \rho K_q$ shots per entity type. We keep track a table to record the current count of occurrences of each type of entity types in the task.

**Label Conversion.** In the few-shot setting, the majority region of an document does not follow the *in-task distribution* (ITD) of $\mathcal{E}$. These regions' tokens are treated as either background or the other types of entities from the *out-of-task distribution* (OTD), whose original labels should be arbitrarily converted into 0 label. In addition, we map the original labels of ITD tokens to relative labels. For example, if we use I/O schema, the relative labels should range from label id 0 to label id $(N-1)$.

---

**Algorithm 1** Cross-document Rejection (XDR) Sampling for Few-shot VDER Task Simulation

1: **Require:** $N, K, K_q, \rho, \mathcal{C}_{base}, \mathcal{C}_{novel}, \mathcal{D}$.
2: Randomly sample $N$ entity types from either $\mathcal{C}_{base}$ or $\mathcal{C}_{novel}$ and obtain $\mathcal{E}$.
3: **Initialize:** $S = \emptyset, Q = \emptyset$
4: **Initialize:** $\mathcal{D}^{\mathcal{E}} = \{\mathcal{D}^e | \forall e \in \mathcal{E}\}$ from $\mathcal{D}$.
5: **Initialize:** $N$ integers $train\_count[e] = 0$ for $\forall e \in \mathcal{E}$.
6: **Initialize:** $N$ integers $test\_count[e] = 0$ for $\forall e \in \mathcal{E}$.
7: // Document sampling for $S$
8: **while** $\min_{e \in \mathcal{E}} train\_count[e] < K$ **do**
9:    Find the least frequent entity type in the current task, i.e., $\hat{e} = \text{argmin}_{e \in \mathcal{E}} train\_count[e]$.
10:    Sample a document $(X_j, Y_j)$ from $\mathcal{D}^{\hat{e}}$
11:    Add $(X_j, Y_j)$ to $S$
12:    **for** $e \in \mathcal{E}$ **do**
13:       Remove the selected document from candidate dataset $\mathcal{D}^e \leftarrow \mathcal{D}^e \setminus \{(X, Y)\}$
14:       Update $train\_count[e]$ if $Y$ contains entity type $e$.
15:       **if** $train\_count[e] > \rho K$ **then**
16:          Mask $(train\_count[e] - \rho K)$ instances of type-$e$ by setting token labels to -1
17:       **end if**
18:    **end for**
19: **end while**
20: // Document sampling for $Q$
21: **while** $\min_{e \in \mathcal{E}} test\_count[e] < K_q$ **do**
22:    Find the least frequent entity type in the current task, i.e., $\hat{e} = \text{argmin}_{e \in \mathcal{E}} test\_count[e]$.
23:    Sample a document $(X_j, Y_j)$ from $\mathcal{D}^{\hat{e}}$
24:    Add $(X_j, Y_j)$ to $Q$
25:    **for** $e \in \mathcal{E}$ **do**
26:       Remove the selected document from candidate dataset $\mathcal{D}^e \leftarrow \mathcal{D}^e \setminus \{(X, Y)\}$
27:       Update $test\_count[e]$ if $Y$ contains entity type $e$.
28:       **if** $test\_count[e] > \rho K_q$ **then**
29:          Mask $(test\_count[e] - \rho K_q)$ instances of type-$e$ by setting token labels to -1
30:       **end if**
31:    **end for**
32: **end while**
33: Label conversion for $\forall (X_j, Y_j) \in S \cup Q$.
34: **return:** $\mathcal{T} = \{S, Q, \mathcal{E}\}$

---

### B.3 Dataset Variants

We fix the testing shot as $K_q$=4. We propose two variants of meta-dataset, each of which pay attention to different challenges in few-shot learning. The statistics is summarized in Table 2: **FewVEX(S)** focuses on single-domain receipt un-

derstanding under N-way K-shot setting. The training and testing classes are both from CORD. The goal is to learn domain-invariant meta-parameters. **FewVEX(M)** focuses on learning domain-agnostic meta-parameters from a combination of receipt and form understanding. Receipt and form documents may appear in the same task.

## C Experimental Setups

### C.1 LLM-based Multimodal Encoder

We pre-train the multimodal Transformer on the IIT-CDIP dataset (Harley et al., 2015). It should be noting that this paper does not focus on the pre-training technique. In fact, our framework does not require a well pre-trained encoder, since the meta-learning will further meta-tune the pre-trained encoder to capture the domain knowledge of $P(\mathcal{T})$. Thus, we stop the pre-training until an $81.5\%$ token classification accuracy.

### C.2 Training Parallelism

We employ the episodic training pipeline to learn the meta-parameters from training tasks (i.e., episodes). At each meta-training step, a total of $\tau$ episodes are trained and then validated to obtain the meta-gradients used for updating meta-parameters.

Both meta-training and meta-testing were run in a multi-process manner. Each of our experiments was run on a total of 4 machines and on each machine there are 8 local TPU devices. Since the parameter size of the Transformer-based encoder is large, we use the 8 devices of each machine to train one single episode in parallel. That is saying, at each meta-training step, a total of 4 tasks are used to compute the meta-gradients.

Both the support (train) and query (test) documents in one task are divided and assigned to 8 devices. The prototypes, the nearest neighbors of data points, or the adapted parameters trained on the local support set, are computed on each local device. For validation on the query set, however, we should consider, the scope of the entire task over different local devices. Therefore, we employ Federated Learning techniques (Zinkevich et al., 2010; Pillutla et al., 2022; Chen and Zhang, 2022a; Tian et al., 2022) operating on multiple devices for a distributed within-task adaptation, where we collect the locally adapted parameters (at each inner-loop step) or the prototypes from the 8 devices of a single episode and average their parameters. Specifically, for training parallelism of each episode/task,

there are 4 steps: (1) on each device, we first adapt a model based on the partial support documents located on the device; (2) then, we collect the adapted knowledge from each of the 8 local devices and aggregate them; (3) on each device, we apply the collected adapted knowledge to the partial query documents; (4) the validation loss on the query subset on each devices are collected and we take an average of them.

## C.3 Baselines

There are mainly two families of approaches for Few-shot VDER. (1) **Meta-learning based Approaches.** Our proposed strategies can improve both metric-based and gradient-based meta-learning methods. To validate our arguments, we compare ContrastProtoNet with its metric-based meta-learning baseline ProtoNet (Snell et al., 2017). We compare ANIL+HC with its gradient-based meta-learning baseline ANIL (Raghu et al., 2019), etc. Extending our work to SOTA meta-learning methods could be one of our future works. (2) **Non-Meta-learning based Approaches.** We did not present a comparison with existing Non-Meta-learning based Few-shot VDER techniques (Wang and Shang, 2022; Wang et al., 2023b) due to the following reasons:

- Existing Non-Meta-learning based Few-shot VDER techniques primarily address **document-level** scenarios ("Entity occurrences do NOT vary from one document to another"). In contrast, our paper focuses on the entity level ("Entity occurrences vary from one document to another"). It is thus not fair to compare methods which were designed under dissimilar problem settings.

- We have conducted comparative experiments by applying (Wang and Shang, 2022) to our BERT-based LLM, a Non-Meta-learning based Few-shot multimodal NER technique, to our specific problem setting. With the same fine-tuning steps (T=15) and learning rate, the F1 results of (Wang and Shang, 2022) on the 4-way 4-shot setting is only **0.115**, while the F1 results of all gradient-based meta-learning methods (including our proposed method) is **over 0.5** and that of all metric-based methods (including our proposed method) is **over 0.23**. However, it may be not fair to compare with (Wang and Shang, 2022) since our paper studies a different setup. Despite this nuance,

we intend to incorporate these results into the revised version of our paper.

## C.4 Hyperparameters

We summarize the hyperparameters in Table 5.

| Hyperparameters | Value |
|---|---|
| $\rho$ | 3 |
| $\gamma$ | 0.6 |
| $U$ | 20 |
| $K_q$ | 4 |

Table 5: Hyperparameters.

# D Evaluation Methods

## D.1 Quantitative Metrics

We consider two types of quantitative metrics.

**Overall Performance.** Following (Park and Kim, 2020; Xu et al., 2020), we use the precision (**P**), recall (**R**) and micro **F1**-score over meta-testing tasks. We use the I/O tagging schema and the "seqeval" (Nakayama, 2018) tool to compute the P/R/F1.

**Task Specificity (TS).** In the proposed framework, we solve out-of-distribution (OOD) detection as a subtask to improve task personalization and avoid spurious features. We calculate a ITD score for each data point representing how likely it belongs to the task-specific distribution. To evaluate how well the learned meta-learners can distinguish in-task distribution (ITD) from the out-of-task distribution (OTD), we calculate AUROC (Xiao et al., 2020) using the ITD scores over all test episodes. A higher AUROC value indicate better TS performance, and a random guessing detector corresponds to an AUROC of 50%. We use the "sklearn.metrics" (Varoquaux et al., 2015) tool to compute the AUROC and plot ROC curves.

## D.2 Visualization

To visualize the TS, we plot the **ROC curves** of all the meta-testing tasks, where each curve represent one task. Another visualization for TS is to show how **ITD and OOD** are distinguished against each other. We randomly select a testing task and exploit **tSNE** (van der Maaten and Hinton, 2008) to visualize the learned embeddings of all the tokens in the task, where ITD tokens are denoted as red points and OTD tokens are blue points.

Furthermore, we use tSNE to visualize the learned embeddings of only the **ITD** token instances in the task, where different colors represent different entity types.

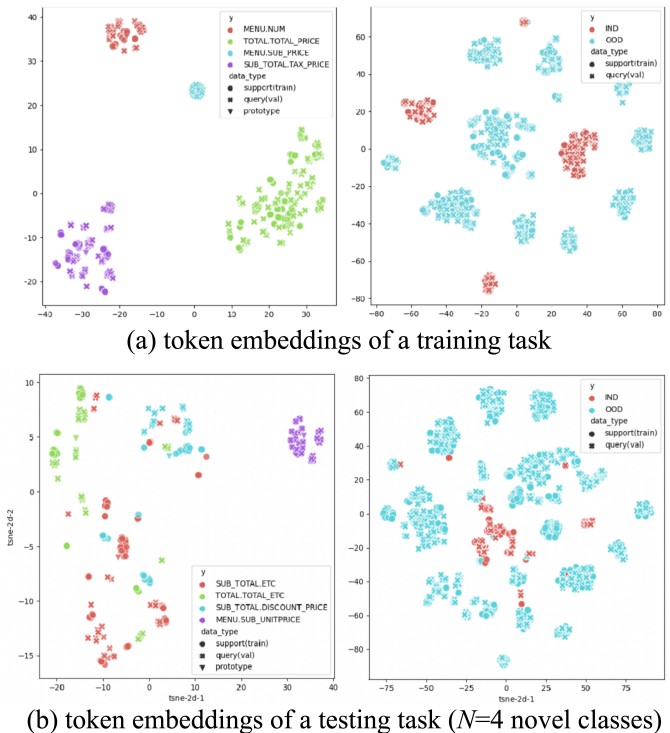

(a) token embeddings of a training task

(b) token embeddings of a testing task (*N*=4 novel classes)

Figure 4: Learned class distribution of a training task and a testing task of 4-way 4-shot setting. The meta-parameters are trained using ContrastProtoNet on FewVEX(S). Solid points represent train (support) tokens, cross points represent val/test (query) tokens, and the triangle points represent prototypes.

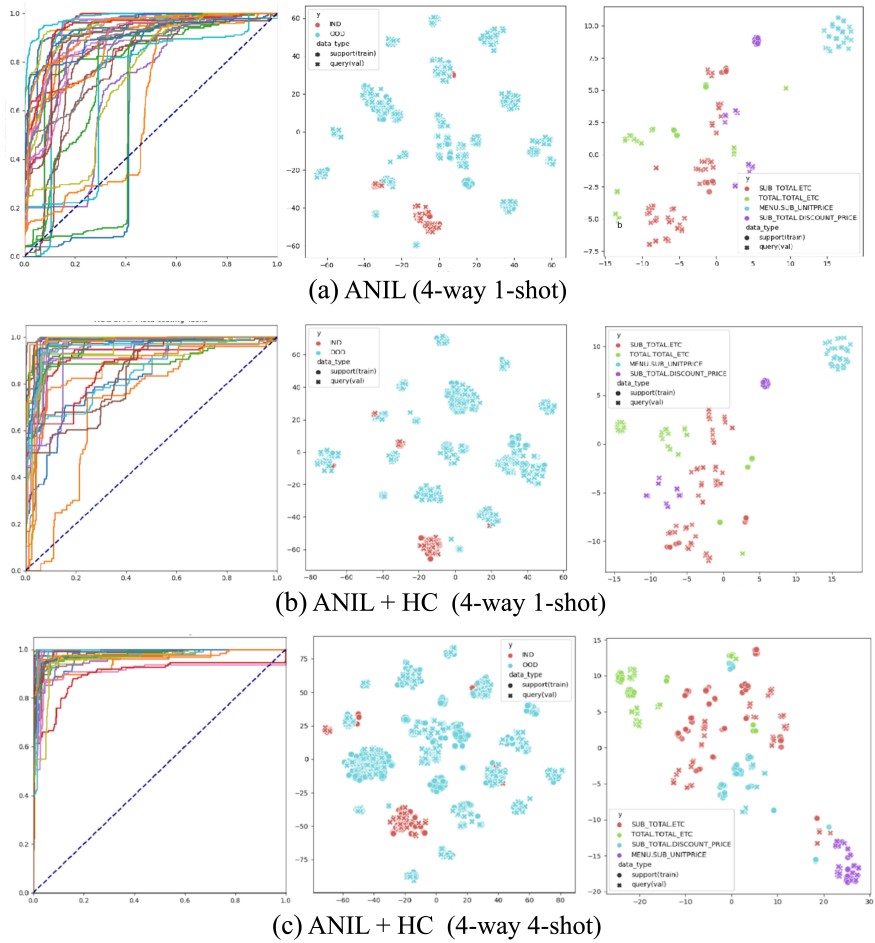

(a) ANIL (4-way 1-shot)

(b) ANIL + HC  (4-way 1-shot)

(c) ANIL + HC  (4-way 4-shot)

Figure 5: Visualization under 4-way 4-shot and 4-way 1-shot settings of FewVEX(S), for ANIL and ANIL+HC.

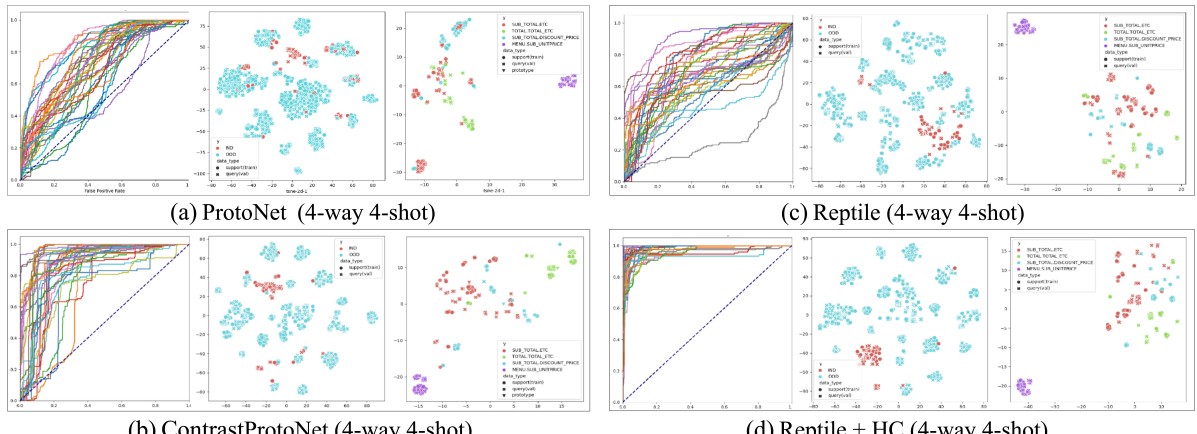

(a) ProtoNet  (4-way 4-shot)

(c) Reptile (4-way 4-shot)

(b) ContrastProtoNet (4-way 4-shot)

(d) Reptile + HC (4-way 4-shot)

Figure 6: Visualization and ROC curves of different methods on the 4-way 4-shot setting of FewVEX(S). For each method, the left subfigure is the tSNE visualization of the learned embeddings of in-task distribution (ITD) entities of a randomly chosen meta-testing task, where different colors indicate different entity types); the middle subfigure shows the tSNE visualization of the learned embeddings of all tokens in the same meta-testing task, where the ITD entities are represented as red points and the out-of-task distribution (OTD) entities or background are represented as blue points; the right subfigure shows the ROC curves of all meta-testing tasks, where each colored line corresponds to one task, representing how ITD is distinguished from OTD based on the model's output logits.

# E   Additional Results

We present additional visualization results in Figure 6, Figure 4, and Figure 5.