# OpenReview forum: "On Task-personalized Multimodal Few-shot Learning for Visually-rich Document Entity Retrieval"
_EMNLP/2023/Conference — EMNLP 2023 Findings_

### Official Review · Reviewer_1N1p · 2023-07-31

**Soundness:** 4

**Excitement:**

3: Ambivalent: It has merits (e.g., it reports state-of-the-art results, the idea is nice), but there are key weaknesses (e.g., it describes incremental work), and it can significantly benefit from another round of revision. However, I won't object to accepting it if my co-reviewers champion it.

**Paper Topic And Main Contributions:**

The paper proposes an approach for the extraction of entities from visually rich documents.

The main contributions of this paper are:
1. Proposing a new formulation of the Few-shot Visually-rich Document Entity Retrieval (FVDER) problem as an entity-level, N-way soft-K-shot learning under the framework of meta-learning.
2. Introducing a new dataset that is designed to reflect practical problems in FVDER.
3. Exploiting a wide range of approaches, including metrics-based and gradient-based meta-learning methods, along with a few new techniques for this new setting.
4. Achieving major improvements over the baselines for FVDER.
5. Identifying potential directions for further improvement, including a better algorithm that distinguishes between the OTD and ITD and a formulation that considers the correlations between entity instances within each meta-learning task.

**Reasons To Accept:**

- the paper presents a novel approach to the challenging problem of Few-shot Visually-rich Document Entity Retrieval (FVDER) and provides significant improvements over the existing baselines
- the proposed methods achieve major improvements over the baselines for FVDER
- the paper also identifies potential directions for further improvement, which can inspire future research in this area
- the proposed dataset that the authors introduce can help the community benchmark new approaches on the proposed task

**Reasons To Reject:**

- the derived dataset is based on the current open-source ones for document understanding, which are small in their size and have a very limited amount of classes.  A dedicated dataset that is built specifically for the purpose of studying few-shot learning for document entity retrieval is needed and it would benefit from a larger number of entities
- the paper does not provide a comprehensive analysis of the computational complexity of the proposed methods, which could be a concern for practical applications.


**Reproducibility:**

2: Would be hard pressed to reproduce the results. The contribution depends on data that are simply not available outside the author's institution or consortium; not enough details are provided.

**Reviewer Confidence:**

1: Not my area, or paper was hard for me to understand. My evaluation is just an educated guess.

**Typos Grammar Style And Presentation Improvements:**

- missing reference on Federated Learning on line 485

---

> ### Author Rebuttal · Authors · 2023-08-28
>
> ### ***Weakness 1: the derived dataset is based on the current open-source ones for document understanding, which are small in their size and have a very limited amount of classes. A dedicated dataset that is built specifically for the purpose of studying few-shot learning for document entity retrieval is needed and it would benefit from a larger number of entities.***
>
> ***Response:***
> We appreciate the comments from the reviewer! We understand the importance of a larger-scale dataset for demonstrating the effectiveness of our method more comprehensively. While acknowledging its significance, we opted to use the derived FewVEX dataset for experiments for the following reasons:
> - ***Sufficiency of Current Dataset to Our Objective***: As we are mostly studying the occurrence shift of entities across documents (as demonstrated in the two examples we shared with other reviews), the most important aspect of the dataset is to formulate the entity-document distributions that can authentically simulate real-world entity-level few-shot situations, where each task needs only a few labeled classes and documents. From that perspective, a dataset built from the existing document understanding ones should be enough to generate a multitude of distinct entity-level few-shot simulations/tasks to effectively evaluate the few-shot learning methods.
> - ***Algorithmic Potential for Future Enhancement:*** While the current dataset is enough to validate our algorithms, we concur that further improvement on increasing the number of entity types and documents will better benefit this research community. In our paper, we made this future enhancement possible by proposing the **Algorithm 1 (in Appendix B on Page 12)**, which facilitate the auto-generation of a large amount of distinct task simulations from open-source document understanding ones with a large amount of classes. We currently select CORD and FUNSD as the input source of Algorithm 1 due to their relatively larger number of classes rather than other available document understanding datasets. If, in the future, document understanding datasets encompassing a wider array of entity types become available, our Algorithm 1 can help quickly derive few-shot datasets with a more extensive number of classes and documents. Using the method proposed in the paper helps to reduce the amount of time used to generate data for few-shot learning cases.
> - ***Established Practice in Few-Shot Learning:*** Deriving few-shot datasets from other non-few-shot sources is well-established in the image few-shot learning domain. Notably, MiniImageNet for few-shot classification emerged from the ImageNet dataset [1].
>
>
>
> [1] Vinyals, Oriol, Blundell, Charles, Lillicrap, Tim, Wierstra, Daan, et al. Matching networks for one shot learning. In Neural Information Processing Systems (NIPS), 2016
>
> ### ***Weakness 2: the paper does not provide a comprehensive analysis of the computational complexity of the proposed methods, which could be a concern for practical applications.***
>
> ***Response:*** We appreciate the comments from the reviewer. We present the computational complexity analysis for both the baseline methods and our proposed approach as follows.
>
> |                                  | Total Parameters | Outer-loop Parameters | Inner-loop Parameters | Inner-loop Space Complexity | Inference Time Complexity | 4-way 4-shot F1 Performance (T=15) |
> |-------------------------|:------------------:|:--------------------------:|:---------------------------|:---------------------:|:-------------------------------:|:-------------------------------------------:|
> |     Fine Tuning  [2]      |    11.1M             |   0                                |                  11.1M         |            O(N)            |          O(T)            | 0.115 |
> |         Reptile         |    11.1M   |  11.1M  |  11.1M | O(N)  |  O(T) | 0.51 |
> |    **Reptile+HC (ours)**  |    11.2M   |  11.2M  |  11.2M | O(N+2)|  O(T) |**0.64**|
> |                  ANIL                    |    11.1M   |  11.1M  |   102k | O(N)  |  O(T) | 0.50 |
> |     **ANIL+HC (ours)**           |    11.2M   |  11.2M  |   200k | O(N+2)|  O(T) |**0.51**|
> |       Vanilla ProtoNet              |    11.0M   |  11.0M  |    0   | O(DL) |  O(1) | 0.03 |
> |       ProtoNet + EOD              |    11.0M   |  11.0M  |    0   | O(DNK)|  O(1) | 0.23 |
> | **ContrastProtoNet (ours)**  |    11.0M   |  11.0M  |    0   | O(DL) |  O(1) |**0.60**|
>
>
> In the above table, N represents the number of classes in a task, and K is the number of labeled entity occurrences per class. L denotes the number of tokens in a document, set as L=320. D signifies the encoded parameter dimension, with D=768. T corresponds to the number of fine-tuning steps in the inner loop. The number of steps has been preset for all gradient-based methods to observe their convergence speed. Our method achieves superior results with the same time complexity and a marginal (0.01%) increase in parameter size. The primary memory cost (11M) is attributed to the BERT-like encoder.
> [2] Zilong Wang and Jingbo Shang. 2022. Towards few-shot entity recognition in document images: A label-aware sequence-to-sequence framework. arXiv preprint arXiv:2204.05819.
>
> ### ***Others: missing reference on Federated Learning on line 485.***
> Thank you for bringing the missing reference to our attention. The federated learning technique we employed is FedAvg [2]. We will correct this in the revised version.
>
> [2] McMahan, Brendan, et al. "Communication-efficient learning of deep networks from decentralized data." Artificial intelligence and statistics. PMLR, 2017.

---

### Official Review · Reviewer_BXH7 · 2023-08-04

**Soundness:** 3

**Excitement:**

3: Ambivalent: It has merits (e.g., it reports state-of-the-art results, the idea is nice), but there are key weaknesses (e.g., it describes incremental work), and it can significantly benefit from another round of revision. However, I won't object to accepting it if my co-reviewers champion it.

**Paper Topic And Main Contributions:**

The paper proposes to solve the problem of entity extraction from visually rich document for unseen entity types. The paper proposes a new dataset for this problem . The solution provided by the paper for this problem consists of three parts 1) multimodal encoder that encodes the input document into a 'task-dependent embedding space' 2) A decoder that does the token labelling 3) a meta learner that learns parameters for the encoder and decoder such that labelling can be done on the out-of-task distribution as well. They propose a few techniques for the meta learning part.

**Questions For The Authors:**

Please provide more clarity with respect to my Point 1 of previous section .

**Reasons To Accept:**

1. Paper deals with the multimodal few shot entity recognition problem which is very real and practical problem
2. Sound experiments have been done among the meta learning approaches proposed and other meta learning approaches .
3. Results are good improvement over the methods compared.

**Reasons To Reject:**

1. Task definition needs better clarity. The paper claims to solve a niche problem which has no dataset but is not clear what the niche is.
 Some of the claims made by the paper about the 'uniqueneness' of the problem are :
Claim by paper : Line 071- Entity occurences vary  dramatically from one document to another
Confusion : Is it not true that in any NER case in real word this happens ?
Claim 2 by paper  : N way means S&Q do not contain any entity types other than E(epsilon)
Confusion : Again is that not true in any NER task ?
Similar claims are repeated in Line 183-200 again with the same set of confusions. I am finding it hard to differentiate these claims of uniqueness from a 'Few shot multimodal NER' setting.
2.Comparisons are neeed against Few shot multiomdal NER techniques , not just meta learning based ones. particularly if the previous point is valid.

**Reproducibility:**

4: Could mostly reproduce the results, but there may be some variation because of sample variance or minor variations in their interpretation of the protocol or method.

**Reviewer Confidence:**

4: Quite sure. I tried to check the important points carefully. It's unlikely, though conceivable, that I missed something that should affect my ratings.

---

> ### Author Rebuttal · Authors · 2023-08-28
>
> ### ***Confusion 1 about Task Definition: A claim by paper is "Line 071- Entity occurrences vary dramatically from one document to another", and the confusion is that “Is it not true that in any NER case in the real world this happens?***
>
> **Response**:
>
> According to our observations, two distinct situations can exist in real-world document image entity retrieval cases:
>
> - **Situation 1: “entity occurrences do NOT vary from one document to another”**. For instance, in three document images, X1, X2, and X3, and three entity types, E1, E2, and E3, this situation assumes that E1 appears once in X1, E2 appears once in X1, and E3 also appears once in X1. Similarly, E1 appears once in X2, E2 appears once in X2, and E3 also appears once in X2. We use a Table below to summarize this example. This scenario is limited to specific document types whose layout structures and  entity types are predefined, such as Tax forms, where the “custom name” and “custom address” occur in all forms around the same position.
>
>    |                                   |  #Occurrence in X1   |   #Occurrence in X2     |   #Occurrence in X3   |
>    |-------------------------|:----:|:---------:|:------:|
>    |        Entity Type  E1                     |    1  |   1        |     1  |
>    |        Entity Type  E2                     |    1  |   1        |     1  |
>    |        Entity Type  E3                     |    1  |   1        |     1  |
>
> - ***Situation 2: “entity occurrences vary dramatically from one document to another”***. In this case, entity occurrences differ significantly between documents. For instance, E1 occurs 3 times in X1, while E2 appears once and E3 is absent. In contrast, E1 might not be present in X2, E2 might occur 3 times in document X2, E3 appears 2 times in X2.... This situation is often encountered in scenarios with substantial shifts in document layout or content, as seen in datasets like the receipt dataset (e.g., CORD).
>
>    |                                   |  #Occurrence in X1   |   #Occurrence in X2     |   #Occurrence in X3   |
>    |-------------------------|:----:|:---------:|:------:|
>    |        Entity Type  E1                     |    3  |   1        |     0  |
>    |        Entity Type  E2                     |    0  |   3        |     2  |
>    |        Entity Type  E3                     |    1  |   0        |     3  |
>
> To clarify, our claim is that ***“Both situations can happen in the real world. While the first situation has been addressed in prior research [1,2], the second situation has never received research attention.”*** As such, our paper proposes a solution for the second situation.
>
> Furthermore, despite our primary focus on the second situation and our experiments within this context, it's important to note that our method is also applicable to the first situation without necessitating modifications.
>
> [1] Zilong Wang and Jingbo Shang. 2022. Towards few-shot entity recognition in document images: A label-aware sequence-to-sequence framework. arXiv preprint arXiv:2204.05819.
>
> [2] Zifeng Wang, Zizhao Zhang, Jacob Devlin, Chen-Yu Lee, Guolong Su, Hao Zhang, Jennifer Dy, Vincent Perot, and Tomas Pfister. 2022. Queryform: A simple zero-shot form entity query framework. arXiv preprint arXiv:2211.07730.
>
> ### ***Confusion 2 about Task Definition: a claim by paper is "$N$ way means $S$&$Q$ do not contain any entity types other than $\mathcal{E}$", and the confusion is that “Again is that not true in any NER task ? I am finding it hard to differentiate these claims of uniqueness from a 'Few shot multimodal NER' setting.***
>
> **Response**:
>
> Thanks for your question! We want to clarify that we are not claiming that the $S$ & $Q$ documents do not contain any entity types other than the N types in $\mathcal{E}$.  Our claim is that ***“$S$ & $Q$ documents do not contain any LABELED entity types other than the N types in E. Other entity types aside from $\mathcal{E}$ can indeed exist within $S$ and $Q$ documents, but they remain unlabeled and are treated as background or out-of-distribution information.”***
>
> To be more clear, let's provide an illustrative example of the content within an N-way task $\mathcal{T}=(S,Q,\mathcal{E})$.
> - Around 5% tokens in $S$ & $Q$ documents are associated with either LABELED or UNLABELED entity types from the set $\mathcal{E}$, comprising the entities that align with the **in-task distribution (ITD)**.
> - Roughly 20% tokens in $S$ & $Q$ documents belong to UNLABELED entity types **other than $\mathcal{E}$**, which are the **out-of-task-distribution** entities.
> - Approximately 75% tokens in $S$ & $Q$ document belong to UNLABELED background information, also **not part of $\mathcal{E}$** and from out-of-task distribution.
> It's important to note that for these UNLABELED tokens, we have no knowledge about whether they represent background, in-task distribution, or out-of-task distribution entities. More detailed clarification regarding the task information is offered in lines 333-349.  We intend to enhance the clarity of this explanation in the upcoming revised version of the task definition.
>
> Furthermore, it is true that some real-world applications can directly lead to our claim: ***“$S$ & $Q$ documents do not contain any LABELED entity types other than the N types in E. Other entity types aside from $\mathcal{E}$ can indeed exist within $S$ and $Q$ documents, but they remain unlabeled and are treated as background or out-of-distribution information.”***  To illustrate, consider a scenario where a document encompasses five entity types: E1, E2, E3, E4, and E5. Suppose there is a user of the system, who is only interested in 3 of these entities (i.e., E1, E3, E4) and provides few-shot labels for them. The user is **not interested in E2 and E5 and does not label them, but E2 and E5 still exist in the training and testing documents and are left as the background context to the user**. Such a scenario is practical in the realm of Few-shot VNER systems, and our claim aligns with the personalization of entity types for individual tasks/users.
>
>
> ### ***Weakness 2: Comparisons are needed against Few shot multimodal NER techniques , not just meta learning based ones. particularly if the previous point is valid.***
>
> **Response**:
>
> In our paper, we do not present a comparison with existing Non-Meta-learning based Few-shot multimodal NER techniques due to the following reasons:
> - Existing Non-Meta-learning based Few-shot multimodal NER techniques primarily address document-level scenarios ("Entity occurrences do NOT vary from one document to another"). In contrast, our paper focuses on the entity level ("Entity occurrences vary from one document to another").  It is thus not fair to compare methods which were designed under dissimilar problem settings.
> - We have conducted comparative experiments by applying [1], a Non-Meta-learning based Few-shot multimodal NER technique, to our specific problem setting. With the same fine-tuning steps (T=15) and learning rate, the F1 results of [1] on the 4-way 4-shot setting is only **0.115**, while the F1 results of all gradient-based meta-learning methods (including our proposed method) is **over 0.5** and that of all metric-based methods (including our proposed method) is **over 0.23**. However, it may be not fair to compare with [1] since our paper studies a different setup. Despite this nuance, we intend to incorporate these results into the revised version of our paper.

---

### Official Review · Reviewer_KbGr · 2023-08-09

**Soundness:** 3

**Excitement:**

3: Ambivalent: It has merits (e.g., it reports state-of-the-art results, the idea is nice), but there are key weaknesses (e.g., it describes incremental work), and it can significantly benefit from another round of revision. However, I won't object to accepting it if my co-reviewers champion it.

**Paper Topic And Main Contributions:**

The paper addresses the limitations of existing methods by focusing on the under-explored entity-level imbalanced few-shot VDER scenarios. The proposed across-document rejection (ADR) sampling algorithm efficiently handles the imbalanced nature of data by identifying and rejecting irrelevant instances. The use of meta-learning-based approaches, such as HC and ContrastProtoNet, in out-of-task-distribution document contexts enables robust few-shot entity training. The introduction of the FewVEX dataset serves as a valuable resource for evaluating the proposed approaches.

**Reasons To Accept:**

1. The paper addresses a significant problem in the field of industrial NLP applications and provides a novel approach for visually-rich document entity retrieval.

2. The incorporation of asymmetric across-document rejection (ADR) sampling helps to handle the imbalance in entity occurrences in a more realistic manner.

3. The utilization of meta-learning techniques in out-of-task-distribution document contexts enhances the generalization ability of the models and improves upon popular meta-learning baselines.

4. The introduction of the FewVEX dataset not only facilitates the evaluation of the proposed approaches but also sets a benchmark for future research in this area.

**Reasons To Reject:**

1. While the proposed approach is evaluated on the FewVEX dataset, more extensive experimentation on other publicly available datasets could further validate the effectiveness and generalizability of the proposed methods.

2. The paper could provide more detailed explanations or examples of the entity-level imbalanced few-shot VDER scenarios and how they differ from the document-level imbalance. This would help readers better understand the specific challenges posed by entity-level imbalance.

3. Additionally, the paper could provide more specific details on the implementation and experimental setup to ensure reproducibility of the results.

4. The paper could consider discussing the potential biases or limitations in the FewVEX dataset and how they may impact the generalizability of the results. Addressing any potential dataset limitations or biases would enhance the credibility of the experimental findings.

**Reproducibility:**

4: Could mostly reproduce the results, but there may be some variation because of sample variance or minor variations in their interpretation of the protocol or method.

**Reviewer Confidence:**

3: Pretty sure, but there's a chance I missed something. Although I have a good feel for this area in general, I did not carefully check the paper's details, e.g., the math, experimental design, or novelty.

---

> ### Author Rebuttal · Authors · 2023-08-28
>
> ### ***Weakness 1:  While the proposed approach is evaluated on the FewVEX dataset, more extensive experimentation on other publicly available datasets could further validate the effectiveness and generalizability of the proposed methods.***
>
> Thank you for your valuable suggestions! The notion of comparing with other public datasets holds immense potential in substantiating the efficacy and applicability of our proposed methods. However, based on our investigation, there is currently **no public available dataset** specifically designed for entity-level few-shot document entity retrieval tasks, since we are the first studying such real-world tasks.
>
> The FewVex is proposed by our paper, by deriving from existing document understanding datasets (CORD and FUNDS). Our choice to focus on CORD and FUNDS is due to their significant diversity of entity types, which renders them ideal candidates for constructing FewVex.   Conversely, other document understanding datasets were not selected due to their lack of substantial entity type diversity, making them less suitable for the construction of FewVex.  **The proposed FewVEX will be the first public dataset** for entity-level few-shot document entity retrieval. We will release the code and our datasets upon the paper's acceptance. We hope FewVex will emerge as a valuable data source, aiding future researchers within this domain.
>
> Furthermore, we present **Algorithm 1 (available in the Appendix on page 12)**, an **automated dataset construction** algorithm capable of generating novel few-shot datasets with task-specific entity types from existing document understanding datasets with global entity types. The code for this algorithm will be made publicly available, and we anticipate that if, in the future, document understanding datasets encompassing a wider array of entity types become available, our algorithm will facilitate the creation of an even more comprehensive collection of publicly accessible datasets within this context.
>
> ### ***Weakness 2: The paper could provide more detailed explanations or examples of the entity-level imbalanced few-shot VDER scenarios and how they differ from the document-level balanced ones. This would help readers better understand the specific challenges posed by entity-level imbalance.***
>
> We present an illustrative example to compare the scenarios of entity-level imbalanced few-shot VDER and how they contrast with document-level balanced few-shot VDER scenarios:
> - “Document-level balanced few-shot VDER scenarios'”. For instance, in three document images, X1, X2, and X3, and three entity types, E1, E2, and E3, this situation assumes that E1 appears once in X1, E2 appears once in X1, and E3 also appears once in X1. Similarly, E1 appears once in X2, E2 appears once in X2, and E3 also appears once in X2. We use a Table below to summarize this example. This scenario is limited to specific document types whose layout structures and  entity types are predefined, such as Tax forms, where the “custom name” and “custom address' ' occur in all forms around the same position.
>
>    |                                   |  #Occurrence in X1   |   #Occurrence in X2     |   #Occurrence in X3   |
>    |-------------------------|:----:|:---------:|:------:|
>    |        Entity Type  E1                     |    1  |   1        |     1  |
>    |        Entity Type  E2                     |    1  |   1        |     1  |
>    |        Entity Type  E3                     |    1  |   1        |     1  |
>
> - “Entity-level imbalanced few-shot VDER scenarios”.  In this case, entity occurrences differ significantly between documents. For instance, E1 occurs 3 times in X1, while E2 appears once and E3 is absent. In contrast, E1 might not be present in X2, E2 might occur 3 times in document X2, E3 appears 2 times in X2.... This situation is often encountered in scenarios with substantial shifts in document layout or content, as seen in datasets like the receipt dataset (e.g., CORD).
>
>    |                                   |  #Occurrence in X1   |   #Occurrence in X2     |   #Occurrence in X3   |
>    |-------------------------|:----:|:---------:|:------:|
>    |        Entity Type  E1                     |    3  |   1        |     0  |
>    |        Entity Type  E2                     |    0  |   3        |     2  |
>    |        Entity Type  E3                     |    1  |   0        |     3  |
>
>
> ### ***Weakness 3:  The paper could consider discussing the potential biases or limitations in the FewVEX dataset and how they may impact the generalizability of the results. Addressing any potential dataset limitations or biases would enhance the credibility of the experimental findings.***
>
> Thank you for the valuable suggestion regarding the discussion of limitations in the FewVEX dataset. We identify two distinct limitations:
> This is a very good suggestion discussing the limitations in the FewVEX dataset. We think there are two Limitations: (1) An inherent limitation pertains to the *number of entity types* in the FewVEX dataset. The current scope of classes  in the FewVEX may not be extensive. However, we have limited options when choosing document understanding datasets to generate FewVEX, and we decided to use CORD and FUNDS due to their relatively larger number of classes rather than other available document understanding datasets.  Despite such restriction, we introduce Algorithm 1, which provides a solution that automatically generates a few-shot dataset, and if, in the future, document understanding datasets encompassing a wider array of entity types become available, our  Algorithm 1 can help quickly derive few-shot datasets with more extensive classes and will improve the generalization of the proposed system. (2) *Limited Number of Domains*: Another limitation arises from the inclusion of only two domains–receipts and forms—within the FewVEX dataset. The inherent challenge posed by the limited domain diversity is that the available domains may not be sufficient to comprehensively assess the capacity of methods to handle domain shifts. Recognizing this concern, we are committed to expanding the dataset to incorporate additional domains in our future endeavors.
>
> ### ***Weakness 4:  The paper could provide more specific details on the implementation and experimental setup to ensure reproducibility of the results.***
>
> We implemented the baselines and proposed method using Jax and Tensorflow and conducted all experiments across 8 TPU devices. Each experiment was iterated 5 times with distinct random seeds, and the average results were recorded. In terms of model configuration, our approach involves a BERT-based encoder with an output dimension of 768. Subsequently, for gradient-based methods, a 2-layer MLP functions as the decoder.
> For a more comprehensive understanding of our training specifics, we offer additional training-related experimental setup in Appendix C. To enhance reproducibility, we will release our code, thus facilitating the replication of our experiments.

---

### Meta-Review · Area_Chair_9tTi · 2023-09-22

**Recommendation:** 2

**Metareview:**

This submission studies a very real and practical problem of multimodal few shot entity recognition and handles the imbalance issue at entity-level with a newly proposed method. The soundness of the proposed method is validated by the original experiment results and the additional results provided during the rebuttal.

---

### Decision · Program_Chairs · 2023-10-07

**Decision:**

Accept-Findings

**Comment:**

This submission studies a very real and practical problem of multimodal few shot entity recognition and handles the imbalance issue at entity-level with a newly proposed method. The soundness of the proposed method is validated by the original experiment results and the additional results provided during the rebuttal.